# Caregivers' and nurses' perceptions of the Smart Discharges Program for children with sepsis in Uganda: A qualitative study

Justine Behan[1], Olive Kabajaasi[2], Brooklyn Derksen[3], George Sendegye[4], Brenda Kugumikiriza[2], Clare Komugisha[2], Radhika Sundararajan[5], Shevin T. Jacob[2,6], Nathan Kenya-Mugisha[2]*, Matthew O. Wiens[1,2,7]*

1 Institute for Global Health, BC Children's Hospital and BC Women's Hospital + Health Centre, Vancouver, Canada, 2 Walimu, Kampala, Uganda, 3 College of Nursing, University of Saskatchewan, Saskatoon, Saskatchewan, Canada, 4 Mbarara University of Science and Technology, Mbarara, Uganda, 5 Department of Emergency Medicine, Weill Cornell Center for Global Health, Weill Cornell Medicine, New York, NY, United States of America, 6 Department of Clinical Sciences, Liverpool School of Tropical Medicine, Liverpool, United Kingdom, 7 Department of Anesthesiology, Pharmacology & Therapeutics, University of British Columbia, Vancouver, Canada

☯ These authors contributed equally to this work.
* Matthew.Wiens@bcchr.ca (MOW); kenya@walimu.org (NKM)

**Data Availability Statement:** The data generated and analyzed during the current study, which

## Abstract

Sepsis arises when the body's response to an infection injures its own tissues and organs. Among children hospitalized with suspected sepsis in low-income country settings, mortality rates following discharge are high, similar to mortality rates in hospital. The Smart Discharges Program uses a mobile health (mHealth) platform to identify children at high risk of post-discharge mortality to receive enhanced post-discharge care. This study sought to explore the perceptions and experiences of the caregivers and nurses of children enrolled into the Smart Discharges Program and the program's effect on post-discharge care. We conducted an exploratory qualitative study, which included in-person focus group discussions (FGDs) with 30 caregivers of pediatric patients enrolled in the Smart Discharges Program and individual, semi-structured interviews with eight Smart Discharges Program nurses. The study was carried out at four hospitals in Uganda in 2019. Following thematic analysis, three key themes pertaining to the Smart Discharges program were identified: (1) Facilitators and barriers to follow-up care after discharge; (2) Changed caregiver behavior following discharge; and (3) Increased involvement of male caregivers. Facilitators included telephone/text message reminders, positive nurse-patient relationship, and the complementary aspects of the program. Barriers included resource constraints and negative experiences during post-discharge care seeking. With regards to behavior, when provided with relevant and well-timed information, caregivers reported increased knowledge about post-discharge care and improvements in their ability to care for their child. Enrolment in the Smart Discharges Program also increased male caregiver involvement, increased provision of resources and improved communication within the family and with the healthcare system. The Smart Discharges approach is an impactful strategy to improve pediatric post-discharge care, and similar approaches should be considered to improve the hospital to home transition in similar low-income country settings.

include participant interview and focus group discussion transcripts, are not publicly available due to their potentially directly and indirectly identifiable nature. Furthermore, the open sharing of the qualitative data was not included in the study's approved protocol nor included in the participant consent forms. Principal investigators can work with interested parties to re-analyze any of the original data if there are any queries that are not sufficiently addressed in the manuscript. In such scenarios, data can be accessed through through the Pediatric Sepsis CoLab, where the data are reposited (URL TO BE INSERTED). Requests can be made to Jessica Trawin (jessica.trawin@cw. bc.ca) at the Pediatric Sepsis CoLab.

**Funding:** MOW and NKM received funding (#TTS-1809-1939) for this study from Grand Challenges Canada (GCC). The views expressed are those of the authors and not necessarily those of GCC. RS receives funding from the US National Institutes of Health (R01 MH132440). The funders had no role in study design, data collection and analysis, decision to publish or preparation of the manuscript.

**Competing interests:** All authors declare that they have no conflicts of interest to disclose.

# Introduction

Sepsis arises when the body's response to an infection results in organ dysfunction [1]. In 2017, nearly 49 million incident cases of sepsis were reported globally with half occurring in children under five [2]. The global sepsis burden is disproportionately carried by low- and middle-income countries (LMICs), with close to 3 million deaths annually among children under the age of five, most of which occur in African countries [2].

Among children hospitalized with suspected sepsis in low-income settings, mortality rates following hospital discharge are high, and similar to those seen during the hospital phase of illness [3–8]. Often referred to as post-sepsis syndrome, this period of high vulnerability following discharge is often unknown to many parents and healthcare workers who are also poorly equipped to facilitate a robust recovery within a community environment [9, 10]. Current sepsis guidelines have not sufficiently incorporated post-discharge care, leaving health workers with limited guidance [7]. Recent data suggest that improved post-discharge outcomes can be achieved through educational interventions and community-level referrals for follow-up [11]. Such interventions may form critical components necessary to improve the long-term survival of children with sepsis [12]. However, despite their potential value, financial barriers faced by families and children, such as transportation costs, facility fees, and medications prescribed upon discharge, may make the implementation of such programs difficult [6, 13].

The Smart Discharges Program uses a mobile health (mHealth) platform to identify children at high risk of post-discharge mortality for enhanced post-discharge care. The model used in Smart Discharges predicts risk of future mortality using patient demographics, anthropometric measurements, and clinical indicators collected on admission [14–17]. This innovative program provides an opportunity for front-line health workers in resource-poor environments to identify vulnerable children and to streamline a path for follow-up care during the post-discharge period [17]. Counselling and discharge planning target factors that are known or assumed to contribute to post-discharge mortality, including clinical, social, and system considerations [18]. While all participants receive enhanced discharge counselling, high-risk children receive down-referrals to community health facilities for follow-up, which is detailed in a previous publication [12].

The Smart Discharge Program is currently being evaluated to determine its impact on post-discharge mortality among children admitted with suspected sepsis in Uganda, which will be described in a subsequent manuscript [19]. At this time, the perceptions and experiences of the parents/caregivers and nurses participating in this program have yet to be explored. The caregivers are the key users of the program and the nurses are responsible for the delivery of the program. As such, their experiences are crucial to understanding opportunities for program improvement and future scaling. Therefore, we conducted a qualitative study to explore the caregivers' and nurses' experiences as part of the Smart Discharges Program and its effects on the post-discharge care of children who have been hospitalized with suspected or proven sepsis.

# Methodology

## Design

We conducted an exploratory qualitative study with the goal is understanding the lived experiences of study participants [20, 21]. Data were collected through in-person focus group discussions (FGDs) with caregivers of pediatric patients enrolled in the Smart Discharges Program and individual, semi-structured interviews with Smart Discharges Program nurses who had provided counselling to the caregivers during the child's admission. FGDs and interviews were

conducted in person in private rooms within the pediatric wards at the study sites. The study is reported using the Consolidated Criteria for Reporting Qualitative Research (COREQ) (S1 File) [22].

## Study setting

The study was carried out at four hospitals in Uganda, including three public regional referral hospitals (RRH) and one private-not for profit children's hospital. These were the first four hospitals to implement the Smart Discharges Program in Uganda, which has expanded to further hospitals since the time of data collection.

## Sample selection, recruitment, and consent

We used purposive sampling to identify potential participants for the caregiver FGDs with primary consideration given to ensuring balance between key post-discharge outcomes among their children, including readmission, mortality, as well as attendance and non-attendance at post-discharge follow-up appointments. Caregivers were identified using Smart Discharges study records. They were eligible if their child had been enrolled into the Smart Discharges program and had been discharged at least two months prior to recruitment. Recruitment of caregivers was conducted after two months to allow them to share their experiences with follow up appointments and applying discharge education at home. Nurses were eligible to participate in the in-depth interview if they had worked in the Smart Discharges program for at least two months. FGD participants and nurses were invited to participate through telephone calls from the social scientist (OK). All participants provided written informed consent.

## Data collection

FGDs and in-depth interviews were conducted between August 29, 2019 and October 30, 2019. Three female and one male research assistants (including co-authors GS, BK, CK) who were fluent in the local languages (Runyankore, Luganda, and Lusoga) and had 2–4 years of experience in moderating FGDs conducted the FGDs. They were trained and supervised by a female social scientist (OK; MA Sociology), with relevant working experience in conducting qualitative research in Ugandan health settings. Facilitators of the same sex as the participants moderated the FGDs in the local language. Individual interviews with nurses were conducted by OK in English. A focus group discussion guide (S2 File) and a semi-structured interview guide (S3 File) were used to provide structure and consistency to the discussion/interviews while allowing for novel concepts to be shared. Topics in the interview and FGD guides were developed by senior investigators (MOW, NKM, STJ) who have expertise in the field of pediatric sepsis. The FDG guides focused on experiences at the admitting facility, experiences after discharge, processes involving referral for post-discharge clinical care, and barriers to post-discharge care. Interview guides focused on the experience of providing caregiver education and counselling, and reporting of which programmatic components worked well and did not work well. Nurse interviewees were also asked about their observations of the benefits and challenges of the program to the caregivers. Due to budget and logistical constraints, the FGD and interview guides were not pre-tested prior to use and no repeat interviews were done.

## Data analysis

At the end of each FGD, field notes were written and shared with the study Principal Investigator to review and provide feedback to refine the guides for future use. FGDs and interviews were audio recorded and transcribed directly in English by a professional translator and then

OK reviewed for accuracy and consistency. Using a thematic analysis approach, initial open coding of transcripts was done by two investigators (JB and OK) [20]. Following initial coding, the study team determined that thematic saturation had been reached, as no new themes or subthemes were developed from the transcript data near the end of the coding process. Additionally, the data was rich and provided sufficient information for the research team to gain new knowledge to answer the study's inquiries. Initial codes were categorized both within and between sources. From this, the two investigators (JB and OK) independently created the initial coding framework, deductively using the FGDs/interview guide topics and inductively to identify emergent themes and subthemes. The coding framework was revised and refined through discussion between JB, OK, and MOW. During analysis, data was organized using NVivo version 12.0 (QSR, Massachusetts, United States). After development of the coding framework and initial coding, themes were proposed, and discussed between JB, OK, and MOW who jointly agreed on the study themes and then confirmed full team agreement on the final themes.

## Trustworthiness

Trustworthiness and construct validity were achieved through credibility, reflexivity, dependability, conformability, and transferability [21, 23]. Themes and subthemes were triangulated against the field notes of the research assistant who completed the FGDs to ensure credibility of findings during the analytical process. Transcripts were not returned to participants for review and correction. To ensure reflexivity, two researchers (JB and OK) analyzed the data and discussed their findings to produce the coding framework. They discussed potential biases, such as how their own perceptions, experiences, expectations, and cultural contexts may influence their analysis of the data. To do this they asked questions and discussed concerns with the study team, allowing for a sharing of perspectives [24]. The research team ensured dependability and confirmability by writing detailed notes and conducting regular discussions to review progress, the research process, and study findings. To improve the transferability and relevance of the results, we provided contextual information about the study setting, sample characteristics, and interview guides.

## Ethical considerations

The Makerere University School of Public Health, Research and Ethics Committee (SPH-REC # 691) and the Uganda National Council for Science and Technology (UNCST SS #5047) provided ethics approval for this study. Participants provided written informed consent and confidentiality was emphasized throughout the study process. A compensation of 25,000 Ugandan Shillings (approximately 7 USD) was given to each participant as remuneration for time spent on study-related activities.

## Results

A total of 40 participants were invited to take part in the study; of these, 38 participated, and two caregivers declined citing lack of transport and time (see Tables 1 and 2: Participant Demographics). Four FGDs were conducted, consisting of one with fathers and three with mothers or other female caretakers. Twenty-seven of the caregivers were parents of the children enrolled in the Smart Discharges Program, and three were other family members (one grandmother and two aunts). Eight interviews with nurses were completed, two from each participating facility. Each FGD consisted of 7–8 participants and lasted between 90 and 120 minutes, while individual interviews lasted between 45 and 60 minutes.

**Table 1. Nurse participant demographics (individual interviews).**

| Characteristics | Value |
|---|---|
| # of Nurse Participants | 8 |
| Gender (M/F) | 1/7 |
| Age Range (Y) | 21–33 |
| Hospital (# of nurses) | Mbarara RRH (2); Jinja RRH (2); Masaka RRH (2); HICH (2) |
| Education Level (# of nurses) | Diploma (4); Certificate (4) |

## Thematic analysis

Three key themes were identified: (1) Facilitators and barriers to follow-up care after discharge; (2) Changed caregiver behavior following discharge; and (3) Increased involvement of male caregivers. Within each key theme, multiple subthemes were identified and are summarized in Table 3.

## Facilitators and barriers to follow-up care after discharge

**Facilitators to follow-up care after discharge.**   The three main facilitators to follow-up care after discharge were telephone/short message service (SMS) text reminders, a positive nurse-caregiver/patient relationship, and the complementary aspects of the Smart Discharges Program. Follow up care after discharge was defined as applying the discharge care instructions as applicable to their child and attendance at post-discharge follow up visits.

The caregivers were appreciative of and motivated by SMS referral reminders and follow-up phone calls after their discharge. These SMS messages served as a reminder for and provided encouragement to attend the post-discharge follow up at community level facilities.

"*I finished all the visits because the organization kept on reminding me [via SMS] to take the child to hospital for checkup. This organization of Smart Discharge, they could call me every Tuesday reminding me. [This is] something that encouraged me so I finished all the routines of taking the child to the hospital for checkup*" (MaleCaregiver24).

Nurses reported having more success providing discharge education with caregivers when they had built a positive nursing relationship with the patient and caregiver.

"*Because it begins with trying to create rapport. . .and much as the children have come with different infections, when you create that professional relationship they open up. . . They open up easily and the work becomes easier*" (Nurse3).

Additionally, having a strong nurse-caregiver relationship helped to support better care-seeking behavior after discharge. One nurse shared,

**Table 2. Caregiver participant demographics (FGDs).**

|  | # of Participants | Age Range (Years) |
|---|---|---|
| Female Caregiver FGD #1 | 8 | 22–45 |
| Female Caregiver FGD #2 | 7 | 28–37 |
| Female Caregiver FGD #3 | 8 | 21–48 |
| Male Caregiver FGD | 7 | 22–57 |
| TOTAL | 30 | 21–57 |

**Table 3. Qualitative themes and subthemes.**

| Key Theme | Subtheme |
|---|---|
| 1. Facilitators and barriers to follow up care after discharge | Facilitators<br>• Telephone/SMS reminders<br>• Positive nurse-patient relationship<br>• Complementary aspects of the Smart Discharges program |
|  | Barriers<br>• Lack of resources<br>• Discharge against medical advice<br>• Experiences with untrained/unfriendly healthcare workers |
| 2. Changed caregiver behavior following discharge | Relevancy and timing of counselling/education |
|  | Increased caregiver knowledge |
| 3. Increased involvement of male caregivers | Engagement in care |
|  | Provision of resources |
|  | Improved communication among caretakers |

*"Even after going home, the caregivers keep calling us and telling us how their children are doing and the health centers they have visited. The caregivers now have a positive attitude towards healthcare-seeking than before"* (Nurse1).

Caregivers who were a part of Smart Discharges reported feeling valued and cared for both during the discharge phase of hospitalization as well as after discharge.

Each component of the program, including risk assessment, educational materials, counselling, and post-discharge referral, were described as complementary, and that they "...*work hand in hand*" (Nurse7). A nurse explained, "*[the Smart Discharges Program components] are all complementing each other...they are all supporting one another and none of them can work in isolation*" (Nurse4).

**Barriers to follow up care after discharge.** The key barriers to follow-up care were lack of resources, discharges against medical advice, and experiences with untrained/unfriendly healthcare workers. Lack of resources, which included inadequate financial resources, transportation challenges, and lack of family support, was a prominent barrier to attending follow-up care. Caregivers' financial constraints were described as impeding follow up visit attendance, adequate nutritional support of recovering children, and purchasing of necessary medication and supplies. One nurse shared,

*"The high risk patients...When we tell them to go to the nearest health centres some of them are not sure they will have money to do all the three visits...Sometimes they don't mention this at discharge, but after they have gone home and when we call them they say they did not go for follow up because they did not have money"* (Nurse7).

A caregiver explains this further, *"That day reaches when we don't even have the money for transport. Even my husband fails to provide money, so you say 'I will not go that day'"* (Female-Caregiver10). The nurses highlighted the need to balance the financial demands of having a sick child when there are other children to support at home. It was emphasized that caregivers were doing the best they could with the resources that they have. As another nurse described, *"The issue of poverty is also a big problem. The caregivers are seemingly appreciating the information being given, but them practicing what is being preached to them is a challenge"* (Nurse1).

Financial constraints also contributed to patients and caregivers being discharged against medical advice. When patients and family leave before it is medically advisable to do so, they

miss the opportunity to receive adequate discharge counselling or to receive their follow up appointments. A nurse explained,

> "*Some mothers are admitted when they don't have money to buy food and sometimes they have to buy the drugs as well. So, some will be willing to stay in hospital, but they are worried about what the child will eat. They also see no reason of staying in hospital because they don't have money to buy the prescribed medicine. So even though we take time to convince the mother to stay, she will eventually go [home against medical advice]*" (Nurse8).

Negative encounters with healthcare workers at the community health facilities was a barrier identified by nurses as well as caregivers who had attended follow up appointments. These encounters were reported to be negative because not all healthcare workers had received training on the "downward referral" process and, therefore, did not know the process for assessing the children. As a result, healthcare workers were often unfriendly towards patients and caregivers who were reporting for a follow-up assessment. A nurse explained, "*There is a challenge of caregivers who take their children to nearby referral health centres. They face backlashes from health workers there who bash them for taking children to health facilities when they are not sick*" (Nurse1). This was also validated by the caregivers' perspectives, "*When they were discharging us they told us to take our child to the nearest hospital every week, but when I went there, they shouted at me asking me why am bringing my child to the hospital, yet he is not sick*?" (MaleCaregiver23). When caregivers had an initial negative experience at a community health facility when seeking follow up care, this often impacted future care seeking and the desire to attend subsequent visits.

## Changed caregiver behavior following discharge

Caregivers and nurses reported that the discharge education provided often resulted in changed caregiver behavior, noting that the relevancy and the timing of the counselling was important. When information was relevant and well timed, caregivers' knowledge about post-discharge care of their children increased, which facilitated improved care of their recovering child once they left the hospital.

**Relevancy and timing of counselling/education.** Receptivity of caregivers to discharge education was improved by its perceived relevance and the caregiver's ability to engage with the materials. As a caregiver explained,

> "*This discharge was different from the rest because when you take a child to hospital they tell you that the child has malaria; they bill you, give you medicines and ask you to leave. But during this discharge I was told that the child had malaria and the health workers taught me how to prevent malaria; how to take care of the child so that she does not suffer from malaria again and how it affects the child if she continues getting malaria attacks. They also taught me what do at home*" (FemaleCaregiver2).

The education was directly related with how to care for their vulnerable child and relevant to returning home after discharge.

> "*What we do is that we let them know that a time will come and they will be discharged from hospital and what they need to hold as key is that a patient discharged from hospital is not necessarily completely healed but still on the road to recovery*" (Nurse3).

The importance of caregivers receiving education and understanding how it applies to improving outcomes for their children was validated, "*At least if each health worker would talk*

*to a mother and disseminate information regarding the condition of the baby and how to go about it, I am sure this can reduce on the mortalities in their homes"* (Nurse 6). Another key element to receptivity was caregiver engagement with the educational materials and how the educational messages were conveyed. For example,

*"The [caregivers] are interested in [the educational materials] because it has pictorial illustrations. . .which caregivers gleefully find illustrative of what they are required to do. For instance, going to hospital, what to feed on, the need for the husband to escort the wife to hospital. At least that one they find it exciting"* (Nurse5).

Including pictures that illustrated the discharge instructions was engaging, and was also a tool to overcome literacy barriers. The timing of education was also found to be important. As a nurse explained,

*"They don't know if the child is even getting better, so they tend to be nervous. So, what I do is to find the convenient time when the patient is a bit stable. When there are not many attendants around them, when they are not on phone communicating with their people back home. When they are calm and stable. So, I approach them and talk to them about malaria, what brings about malaria, how it can be prevented at home, and this differs from one child to another depending on how it is presented"* (Nurse3).

By ensuring the caregiver is both physically and mentally in a space where they can learn, they are more able to retain the information.

**Increased caregiver knowledge.** The majority of caregivers shared that at least portions of the education they received from Smart Discharges Program was new information to them. Nurses reported caregivers learning about how to identify danger signs and the critical timing to seek care for their children.

*"Caregivers are at home but they don't know about these danger signs. When we talk to them about these, they appreciate and promise to always take their children to nearby health centres as soon as these danger signs manifest so as to get first aid"* (Nurse7).

One father explained a behavior change he observed in his wife,

*"It wasn't me they found in the hospital. It was my wife and she learnt many things because I saw her changing. She used not to respond quickly in case of any illness but now she rushes to the hospital at a time she sees a sign of illness"* (MaleCaregiver29).

Many nurses and caregivers spoke about how nutritional education was applied following discharge.

*"There are some mothers who come with children in hospital. The child is one-year-old but he/ she is only feeding on breast milk. . . there is now some change. Some have been coming here. . . in our follow up unit. . . And you find that the child has completely changed in terms of weight addition and generally they look healthy. We attribute this to the information we give them. Because we have been emphasizing to them that the child needs to be introduced to other feeds at 6 months of age, so we feel they have embraced this message. We have also reduced post-discharge mortality and reinfection because of our continued teachings about feeding children on a balanced diet which improves their immune systems to fight against infections"* (Nurse1).

Caregivers also explained that the same principles that were learned through Smart Discharges were applied to the families' other children. Examples included, feeding all of their children nutritious food and improved care-seeking behaviours when any child in the family fell sick.

A male caregiver shared his experience,

"*All the things about nutrition, am applying them to my other children and they look healthy. For example, I have chicken, I have fruits like paw-paws* [papaya], *now I give them, I used to sell all of them and use money to drink. Now I repented*" (MaleCaregiver25).

**Increased involvement of male caregivers.**   The programs impact on male involvement in their child's care was seen most prominently through changes in the father's engagement in care, provision of resources for care, and in changes in health-related communication both within the family and with the healthcare system.

**Engagement in care.**   Nurses saw benefits when education/counselling was done with the child's father present.

*"Some mothers are worried that although they are bound to stay in hospital, it is the husband who should have the final say on this. So in such a situation what I do, I go and give health education to the mother in the presence of the husband and during the session I emphasize the importance of staying in hospital and the husband's responsibility. I encourage the husband to financially support the mother for the duration of her stay in hospital and in the end both parties agree to stay in hospital*" (Nurse8).

This example highlights how the Smart Discharges education increased male caregivers' engagement at the facility level. Several caregivers shared examples of how their husbands' engagement in their children's care also increased at home.

"*My husband used not to even give us* [money for] *transport to the hospital when I used to ask him for it. . .but since I joined* [Smart Discharges] *my husband is active. We always come with him to the hospital. He buys food and other responsibilities*" (FemaleCaregiver19).

Furthermore, receiving discharge education has given fathers a means to improve their relationships with their children. "*It has helped me in increasing my relationship with my child, because what I read on the* [Smart Discharges] *card. I buy for her some food like an egg. She becomes so happy*" (MaleCaregiver29).

**Provision of resources.**   Participants reported that fathers demonstrated an improved understanding of the needs of the child and the primary caregiver. This awareness in turn resulted in increased provision of resources for the child and primary caregivers. One caregiver explained,

"*At the time of discharge, it was not me with the child, it was my husband that was there with the child. He learnt how to care for the child. Since then, when I tell him that I want this for the child, he immediately gives it to me because he does not want to go back to the situation that we were in of spending money in the hospital. Every time I tell him that at least the child has to eat eggs twice a week. He makes sure that he buys the eggs*" (FemaleCaregiver21).

Other caregivers shared similar sentiments and acknowledged that after their husbands received discharge education, they bought foods to follow nutritional guidance and purchased

the medications prescribed to their children. Male caregivers were more open to providing transportation support for their children to attend follow up appointments or to return to the hospital when the child was sick again. One male caregiver explained, "*When I reached home I made sure that I buy everything he needs [and] take him to the hospital for checkups until he is totally healed*" (MaleCaregiver27).

**Improved communication among caretakers.**　Participation in the program fostered discussions between fathers and mothers about their child's wellbeing and how to support their vulnerable child. Often the father's cell phone number was provided to receive follow-up reminders, invoking the father to play an important role in communication with the mother.

"*Smart Discharge has helped me, in getting involved in the care of my child because whenever I get a call from Smart Discharge, I go and tell my wife that we have to go back to the hospital so that we can learn on how to take care of our child. Therefore, this increases on my relationship with my wife and the child*" (MaleCaregiver28).

Additionally, improved communication between the healthcare system and male caregivers helped them to better understand why their child was hospitalized and to follow discharge instructions.

"*They are different because previously, they used to just pack for us pills without explaining so much on what you should do. But when we joined Smart Discharge, things changed because before discharging us they first talk to us*" (MaleCaregiver26).

## Discussion

The results of this study improve our understanding of caregivers and nurses experiences in the Smart Discharges program, a child-centered approach to peri-discharge care which seeks to improve outcomes among children hospitalized with sepsis in Uganda. Our qualitative data indicate that this program facilitates outpatient follow-up, improved quality of home-based care practices, and involvement of male caregivers. Resource constraints and negative experiences with local healthcare workers were the most prominent barriers to completing recommended post-discharge follow up and, thus, represents an area of opportunity to further improve post-discharge outcomes.

Though the common paradigm of "if sick, seek care" can work well when caregivers can accurately determine when to seek care, during the post-discharge period children may deteriorate quickly, and early signs of such deterioration may be subtle and difficult for parents to identify [25]. Caregivers may not recognize overt danger signs of severe illness [26]. Our results clearly demonstrate the positive impact of our discharge education program on caretakers' perceived knowledge and confidence in providing appropriate care to their child during the post-discharge period. As such, the transition of care from facility-based care to home-based care by the children's caregiver may be more effective if caregivers have the necessary education to easily recognize danger signs and not to delay seeking care when necessary [26]. Our results also highlight the importance of caregiver engagement for effective knowledge transfer. This information exchange was largely achieved through consistent, clear, and easy to understand materials [27] that were tailored to the local context and for knowledge users who may have low literacy levels.

Participants reported the extension of benefits to other family members and to the broader community. Both female and male caregivers who participated in this study reported applying

the Smart Discharges principles to their other children and sharing them within their communities. Many commented on the economic implications and cost savings when they were able to prevent infections/sepsis and the subsequent hospitalizations/medical care for the whole family. Future work should address how we can strengthen educational engagement concerning the care of clinically vulnerable children at the community level, such as through schools, churches, and other community groups [28].

Within the formal health system, receptivity to routine follow-up of discharged children is critically important to effective long-term management of sepsis and was a noted barrier to post-discharge care. Education for health workers at local health facilities on post-sepsis syndrome may resolve the confusion that healthcare workers face when a seemingly well child comes for follow-up. Healthcare workers will understand that the primary purpose of the post-discharge follow-up visit is to determine if the child is continuing to make a robust recovery or whether any signs or symptoms are present that warrant further care or investigation. Guidance such as the Integrated Community Case Management (iCCM) focuses not merely on the evaluation of illness, but also the evaluation of the "at risk child", and should be updated to include vulnerable children who were recently discharged from hospital [29]. It is critical that the paradigm of sepsis care shifts from a primary focus on acute care to a more holistic focus that does not neglect the critical period of vulnerability following hospital discharge, which has been largely neglected in both policy and practice [6, 30].

When children's male caregivers were included in the Smart Discharges Program, it fostered their involvement in their child's care. Often, the phone number used for SMS follow-up was that of the father. This arrangement automatically created engagement with the fathers that otherwise may not have existed. Engagement in care was reflected in an increased interest in the post-discharge care process as well as increased provision for essentials of care, such as food and transportation. Gopal, *et al.*, highlight that there is a positive association between male involvement and better maternal and child health outcomes in Uganda [31]. They suggest involving male community members, specifically fathers and community leaders who are attuned to the context, to improve the approach to how males are involved in care and to help support sustainable behavior change [31].

Financial constraints were frequently reported as a key barrier to care following discharge, especially with regards to transportation. Many caregivers who participated in this study shared that if they do not have the money necessary for transport, that they will not be able to attend follow up appointments. Financial constraints have been identified as a barrier to follow up care in other health contexts [26] and has prompted a call for further interventions to target patients facing transportation challenges [32]. For caregivers with constrained resources, a more cost-effective alternative to improving post-discharge care, such as the use of telemedicine or community health workers for post-discharge follow-up, could have a substantial impact. When a physical, in-person follow-up is not possible, remote patient assessment via phone or other communication technologies could serve as a viable alternative [33, 34]. The COVID-19 pandemic resulted in the expanded use of telemedicine and other technologies to deliver healthcare services in Uganda [35]. Such experiences should be leveraged so that healthcare workers and patients/caregivers can further optimize the efficiency and quality of post-discharge care [26]. The use of community health workers is widespread in many regions of Africa, but their work is often not well linked to the broader health system with respect to information transfer. Improved digital linkages between community health workers and their referral facilities could substantially improve the provision of post-discharge care through discharge notifications being sent directly to community health workers [36, 37].

## Limitations and strengths

This study is subject to both limitations and strengths. Firstly, the findings of this study were specifically focused on an intervention within the context of a research evaluation. As such, implementation science studies are needed to facilitate integration of this type of program into routine care. Within the study's participants, only one caregiver had experienced the post-discharge death of their child. Therefore, the experiences with the most adverse and serious outcome was not well captured by the sample. Finally, this study was conducted in a single country and further work is needed to understand how this approach may be transferable to other regions. The main strength of this study was hearing the perspectives of the key users of the Smart Discharges Program, which includes both parents and nurses from four different hospitals.

## Conclusions

The Smart Discharges Program is a digitally assisted health innovation that facilitates discharge follow-up and home-based care practices. The Program helped to educate caregivers and better engage fathers to care for their vulnerable child after discharge, and facilitated improved nurse-caregiver relationships. Opportunities to improve pediatric discharge care include developing interventions to address caregivers' resource constraints and negative experiences when seeking follow-up care. The Smart Discharges approach is an impactful strategy to improve pediatric post-discharge care, and similar approaches should be considered to improve the hospital to home transition in other low-resource settings.

## Supporting information

**S1 File. Consolidated criteria for reporting qualitative research (COREQ) checklist.**
(PDF)

**S2 File. Focus group discussion guide for the parents/caregivers.**
(DOCX)

**S3 File. Interview guide for nurses.**
(DOCX)

## Acknowledgments

We would like to acknowledge the WALIMU team, staff at the participating hospitals, and patients and caregivers for their participation.

## Author Contributions

**Conceptualization:** Olive Kabajaasi, Clare Komugisha, Radhika Sundararajan, Shevin T. Jacob, Nathan Kenya-Mugisha, Matthew O. Wiens.

**Data curation:** Olive Kabajaasi, Brooklyn Derksen, George Sendegye, Brenda Kugumikiriza, Clare Komugisha, Matthew O. Wiens.

**Formal analysis:** Justine Behan, Olive Kabajaasi, Brooklyn Derksen, Matthew O. Wiens.

**Funding acquisition:** Shevin T. Jacob, Nathan Kenya-Mugisha, Matthew O. Wiens.

**Investigation:** Olive Kabajaasi, Clare Komugisha, Shevin T. Jacob, Nathan Kenya-Mugisha, Matthew O. Wiens.

**Methodology:** Justine Behan, Olive Kabajaasi, Brooklyn Derksen, Radhika Sundararajan, Shevin T. Jacob, Nathan Kenya-Mugisha, Matthew O. Wiens.

**Project administration:** Olive Kabajaasi, Clare Komugisha, Matthew O. Wiens.

**Resources:** Nathan Kenya-Mugisha, Matthew O. Wiens.

**Supervision:** Olive Kabajaasi, Shevin T. Jacob, Nathan Kenya-Mugisha, Matthew O. Wiens.

**Validation:** Justine Behan, Olive Kabajaasi, Clare Komugisha, Radhika Sundararajan, Shevin T. Jacob, Nathan Kenya-Mugisha, Matthew O. Wiens.

**Writing – original draft:** Justine Behan, Olive Kabajaasi, Matthew O. Wiens.

**Writing – review & editing:** Justine Behan, Olive Kabajaasi, Brooklyn Derksen, George Sendegye, Brenda Kugumikiriza, Clare Komugisha, Radhika Sundararajan, Shevin T. Jacob, Nathan Kenya-Mugisha, Matthew O. Wiens.

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
