## [Decision Letter · Decision Letter 0]

14 May 2024

PONE-D-23-37812Caregivers’ and nurses’ perceptions of the Smart Discharges Program for children with sepsis in Uganda: A qualitative studyPLOS ONE

Dear Dr. Wiens,

Thank you for submitting your manuscript to PLOS ONE. After careful consideration, we feel that it has merit but does not fully meet PLOS ONE’s publication criteria as it currently stands. Therefore, we invite you to submit a revised version of the manuscript that addresses the points raised during the review process.

**ACADEMIC EDITOR: **The manuscript was well-written and the findings are important to support an intervention which would address a major cause of morbidity and mortality among children in Uganda. In view of difficulty to secure a reviewer, I have taken the step to become one of the two reviewers for this manuscript.The past comments and responses to the previous review process under PLOS Global Public Health have been noted and taken into consideration. Please address the minor corrections required for this manuscript. Thank you.

We look forward to receiving your revised manuscript.

Kind regards,

Chai-Eng Tan

Academic Editor

PLOS ONE

Journal Requirements:

3. In the online submission form, you indicated that [The data generated and analyzed during the current study, which include participant interview and focus group discussion transcripts, are not publicly available due to their potentially directly and indirectly identifiable nature. Furthermore, the open sharing of the qualitative data was not included in the study’s approved protocol nor included in the participant consent forms. Principal investigators can work with interested parties to re-analyze any of the original data if there are any queries that are not sufficiently addressed in the manuscript. Reasonable requests can be made to Walimu via corresponding author Dr. Nathan Kenya-Mugisha.]. 

Reviewers' comments:

Reviewer's Responses to Questions

**Comments to the Author**

1. Is the manuscript technically sound, and do the data support the conclusions?

Reviewer #1: Yes

Reviewer #2: Yes

2. Has the statistical analysis been performed appropriately and rigorously? 

Reviewer #1: N/A

Reviewer #2: N/A

3. Have the authors made all data underlying the findings in their manuscript fully available?

Reviewer #1: Yes

Reviewer #2: Yes

4. Is the manuscript presented in an intelligible fashion and written in standard English?

Reviewer #1: Yes

Reviewer #2: Yes

5. Review Comments to the Author

Reviewer #1: A sub theme for key theme (1).... Parents Bounding

You interviewed with a family that their kid had been hospitalized in the Pediatric ward .

They bound to child care and they were interested to stay with child and care her/his.

Reviewer #2: Congratulations on conducting such a useful intervention to address post-discharge care for children with sepsis particularly in a country with limited resources. I have read through the manuscript in detail, including the responses to the previous reviewers. I would like to offer some additional comments to further improve the manuscript further.

Abstract:

The abstract entered into the editorial system has a minor typo error "well-imed information" should be "well-timed information". This typo was not seen in the main text document.

The objective mentioned in the abstract was "to explore the perceptions of the caregivers and nurses..." Please ensure that this corresponds to the main text under the Introduction section, which reads "to explore the caregivers' and nurses' experiences of the Smart Discharges Program and its effects on the post-discharge care." This could be summarised in the abstract to include the term "experiences" and "effects on post-discharge care".

As previously mentioned by past reviewer, it is best to omit the phrase "phenomenological approach" in the abstract.

Table 1: A table to show the total bed capacity and paediatric bed capacity is not necessary.

Data collection: How were the focus groups formed? I take note that one focus group was specifically for male caregivers. What about the other focus groups for women? Was there a mix of participants with higher and lower income groups / education status or were these focus groups formed based on the location of participant recruitment? This information could influence the content of the discussions of each focus group.

Data analysis:

Rather than mentioning code saturation, it would be better to mention that thematic saturation was achieved. This would justify why no further focus groups were conducted.

Trustworthiness

The main purpose of reflexivity is to ensure that personal biases and assumptions are not imposed on the analysis. The sentence "They discussed potential biases, questions, and concerns, allowing for a sharing of perspectives." did not make it explicit enough that the potential biases have been reflected upon and care taken to ensure that they did not influence the analysis of the data. This should be rephrased further.

Table 2b

I have noted your response regarding the previous reviewers' suggestion to include the caregivers' educational status in this table. Would it be possible to include information about the child's age and post-discharge outcomes in the table? This will support your sample selection strategy in the methods section.

Overall the results and discussions were well-written. The issues faced by caregivers who have lower health literacy and the various social determinants of health are well-demonstrated in your findings.

Limitations

I noted that previous reviewers recommended to include lack of piloting of the interview guides as part of the limitations. In qualitative studies, piloting is not a must as the topic guide can be refined with additional interviews or focus group discussions, provided they remain anchored on the main research questions. Hence, I would recommend that this is removed from the limitations section. Rather, focus on issues that affect the credibility, transferability, dependability and confirmability of a qualitative study. line 480: "Within the study participants, only one caregiver experienced post-discharge death of their child" - suggest to add in "only". Again, adverse outcomes do not only refer to death, but also other things such as readmission. Therefore, this information should be in your table 2b.

Do include the strengths of this study as well, besides the limitations.

I hope that these comments would be helpful to the authors.

6. PLOS authors have the option to publish the peer review history of their article (what does this mean?). If published, this will include your full peer review and any attached files.

Reviewer #1: No

Reviewer #2: **Yes: **Chai-Eng Tan

---

## [Author Response · Author response to Decision Letter 0]

27 Jun 2024

Response to Reviewer 1

Reviewer Comment 1: Lines 56–88: The introduction provides a clear overview of the global burden of sepsis among children, especially in LMICs. Consider adding a brief overview of the burden and management of sepsis in Uganda. This will enhance the context of the study.

Response 1: Thank you for the suggestion to incorporate additional detail regarding sepsis management into the introduction. While we agree that such details represent important content, after much consideration we feel that such an addition would detract from the main scope of this paper, which reflect management during/after discharge. However, we have added a sentence into the introduction regarding the fact that sepsis guidelines currently do not incorporate guidance about discharge/post-discharge care. This is a major limitation of current guidelines, and one which needs to be rectified in future iterations of sepsis guidelines for LMIC settings. 

We have added the following sentence:

Current sepsis guidelines have not sufficiently incorporated post-discharge care, leaving health workers with limited guidance.

Reviewer Comment 2: Write the complete sentence when "Smart" is used for the first time (Signs, Medications, Appointments, Results, and Talk with Me), and then you can use the abbreviation in the second use and so on.

Response 2: In the context of “Smart Discharges” the word Smart is not an acronym and does not stand for Signs, Medications, Appointments, Results, and Talk with me.

Comment 3: The authors have identified the lack of exploration of caregivers' and nurses' perspectives as a research gap that this study aimed to address. Line 84: There is a need to expand on why understanding caregivers' (service consumers) and nurses' (service providers) experiences and perceptions is crucial and how their insights could contribute to the improvement of the Smart Discharges Programme’s effectiveness. This will highlight the significance of this qualitative study. Also provide a reference for the Smart discharges protocol.

Response 3: Thank you for this comment, the following has been added to address this (lines 86-90):

At this time, the perceptions and experiences of the parents/caregivers and nurses participating in this program have yet to be explored. The caregivers are the key users of the program and the nurses are responsible for the delivery of the program. As such, their experiences are crucial to understanding opportunities for program improvement and future scaling. 

The following reference has been added to the introduction, which outlines the Smart Discharges protocol: 

Wiens MO, Kumbakumba E, Larson CP, Moschovis PP, Barigye C, Kabakyenga J, et al. Scheduled Follow-Up Referrals and Simple Prevention Kits Including Counseling to Improve Post-Discharge Outcomes Among Children in Uganda: A Proof-of-Concept Study. Global Health: Science and Practice. 2016;4(3):422-34.

Reviewer Comment 4: Generally, the methodology section is well described. However, some parts need to be clarified:

Study setting: The authors should provide a rationale for selecting participants from the four referral hospitals. Are the four hospitals representative of health facilities offering the Smart Discharge Program (SDP) or the broader healthcare system in Uganda? Were there implications for transferability? If so, that should be acknowledged in the limitations section.

Response 4: The following rationale was added as to why participants were selected from the four referral hospitals in this study (lines 104-107):

The study was carried out at four hospitals in Uganda, including three public regional referral hospitals (RRH) and one private-not for profit hospital (Table 1). These were the first four hospitals to implement the Smart Discharges Program in Uganda, which has expanded to further hospitals since the time of data collection. 

The limitations around the transferability of study findings have been added to the limitations section (lines 482-484).

Reviewer Comment 5: Inclusion criteria: Lines 111–114: Please justify the 2-month inclusion criteria for nurses and caregivers – is this the timeframe within which nurses and caregivers are likely to have had significant exposure to the SDP? Also describe how the participants were identified by the Social Scientist (OK). From medical records maintained at the hospitals or SDP records?

Response 5: The rationale for the 2-month timeframe for caregivers to be included in the study and how participants were identified by OK has been added to the methods section (lines 114-118): 

Caregivers were identified using Smart Discharges study records. They were eligible if their child had been enrolled into the Smart Discharges program and had been discharged at least two months prior to recruitment. Recruitment of caregivers was conducted after two months to allow them to share their experiences with follow up appointments and applying discharge education at home.

Reviewer Comment 6: How were the FGD and the semi-structured interview guides developed? Were they based on expert opinions, previous research, existing literature, or standardised tools? 

Response 6: More detail on how the FGD and the semi-structured interview guides were developed has been added to our Methodology under the Data Collection section (lines 132-134):

Topics in the interview and FGD guides were developed by senior investigators (MOW, NKM, STJ) who have expertise in the field of pediatric sepsis.

We also improved the limitations section, to highlight the limitations within the development of the data collection tools (478-480):

Secondly, piloting of data collection tools and use of a conceptual framework were not components of the design of the interview guides, which may have decreased the validity of the instruments used to collect data.

Reviewer Comment 7: Line 174: Include a rationale for conducting separate FGDs for fathers and mothers/other female caregivers. Did the perspectives and experiences differ between the two groups? Why was capturing their unique viewpoints important for the study's objectives? Did this improve the quality of the data collected?

Response 7: In Uganda, fathers and mothers typically play very different roles in the care of children. Thus, it was deemed important to separate father and mothers into different focus groups. Furthermore, to make the discussions more comfortable for participant, especially the women, we decided that separate focus group discussions would be preferred.

Reviewer Comment 8: Lines 131-133 provide an explicit description of how the researchers tracked, monitored, and determined saturation/richness of data across FGDs and interviews. Currently, the description is not adequate. This will help provide justification for the relatively small sample size of 4 FGDs and 8 interviews conducted and underscore reliability of the qualitative findings. 

As an illustration: To track and monitor saturation, the researchers combined sampling, data collection, data analysis and assessment rather than treating them as separate stages in a linear process (Moule, Aveyard & Goodman). The researchers went through multiple sequential analysis rounds until no new information or themes were generated. Saturation was subjectively determined by the researchers to have occurred after conducting four FGDs and eight interviews.

Moule Pam, Aveyard Helen and Goodman Margaret. Nursing Research: An Introduction. 3rd Edition. 2017. Sage Publications. London/Losangeles/New Delhi/Singapore/Washington DC.

Response 8: We agree that our previous description was lacking important details, and have improved our analysis section to justify our sample size. See our Data Analysis Section (lines 143-160).

Reviewer Comment 9: Line 137: The FGD and interview guides were not pre-tested prior to use, and no repeat interviews were done. Provide valid reasons why (QOREC item 17). Was this because of limited participant availability, time/resource allocation, high experience levels of the researchers who designed the guides, etc.? 

Response 9: The FGD and interview guides were not pre-tested prior to use and no repeat interviews were done due to budget and logistical constraints at the time of data collection. This was added to the methods section (lines139-141) and the limits of this were added to the limitations section.

Reviewer Comment 10: Also participant checking was not done. Please provide a valid reason why (QOREC item 28). Participant checking evaluates the credibility of the results by testing the data, analytic categories, interpretations, and conclusions with representatives of the group from whom the data were originally obtained. How was this gap in credibility-checking addressed? 

Response 10: Member checking was not done due to the same budgetary and logistical constraints that limited pre-testing of the FGD and interview guides. For example, we could not reliably follow up with participating caregivers because they reside in distant and rural locations. Validity of study results was maximized through our data analysis process, which involved open coding by two independent coders, followed by inductive and deductive identification of themes and subthemes, and discussion across the study team. These steps are described in detail in lines 143-160.

Reviewer Comment 11: Regarding triangulation of data sources – how was the data from FGDs and interviews used to form the themes? Did the researchers compare, contrast and integrate themes from FGDs and interviews? If so, how were the themes from FGDs and interviews integrated to identify areas of convergence and divergence? This will help the reader to understand the themes and structuring of the study results. 

Response 11: The data analysis section has been updated to answer the above questions (below excerpt from lines 144-160):

Using a thematic analysis approach, initial open coding of transcripts was done by two investigators (JB and OK)… Initial codes were categorized both within and between sources. From this, the two investigators (JB and OK) independently created the initial coding framework, deductively using the FGDs/interview guide topics and inductively to identify emergent themes and subthemes. The coding framework was revised and refined through discussion between JB, OK, and MOW. After development of the coding framework and initial coding, themes were proposed, and discussed between JB, OK, and MOW who jointly agreed on the study themes.

Reviewer Comment 12: The Results section is well organised, featuring clear headings and subheadings. It effectively presents the combined perspectives of SDP service providers and service consumers. However, the authors should pay attention to the following areas:

Lines 178–180: If available, the authors should add more demographic information to Table 2 (e.g., level of education and employment status) to enhance the description of the studied population.

Response 12: Though we do have some of these data for mothers, we do not have these data for nurses and for fathers/male caregivers. While we agree that such data may be useful, we unfortunately cannot report these. Regardless, we do believe that the responses speak for themselves and though we perhaps cannot fully contextualize them form a social and economic perspective, we feel these results remain valuable.

Reviewer Comment 13: In alignment with the descriptive phenomenological design and objectives of the study, it is essential for the authors to ensure a balanced representation of viewpoints and quotes from both caregivers and nurses across all themes and subthemes. E.g.

• Facilitators to follow-up care after discharge: Line 220-221 and 222-226 Consider including quotes from caregivers. 

• Barriers to follow-up care post-discharge: 267-269 Consider providing quotes from caregivers. 

Response 13: Within both the facilitators and barriers to follow up care after discharge sections, quotes from caregivers have been included to ensure that there is balanced representation of viewpoints.

Reviewer Comment 14: Consider including information on caregivers’ and nurses’ recommendations for improvement of SDP in the results section. Files S2 and S3 included questions to elicit participants' perspectives on how to address challenges or barriers and/or improve the SDP. Ref File S2 &S3. 

Response 14: We appreciate the reviewers careful reading of the supplementary files. Though we did capture some information on how to further improve the SDP, we did not feel we had sufficient space in this present manuscript to include it, given its current length. We plan to utilize these data for a separate paper in the future, once we have had a chance to conduct further qualitative work in this area.

Reviewer Comment 15: The discussion effectively summarises the key findings of the study, highlighting the positive impact of the SDP. I strongly recommend that the authors enrich the discussion by providing or referencing evidence-based practices in the approaches they propose to strengthen post-discharge care. Also nurses’ and caregivers’ recommendations for improvement should be considered. E.g. Lines 418–420: community engagement – the authors propose strengthening educational engagement at the community level, such as by collaborating with schools, religious institutions, and community groups. How could these partnerships be formed and sustained? Reference the available practice/literature. 

Response 15: Thank you for this important comment. We agree that such partnerships may be critical in the future success of programs such as ours. However, we have not yet delved into the best strategy to scale building such partnerships. The focus of this paper was not to develop these strategies but rather guide future efforts to strengthen our programs. We are beginning to find ways to do this, but do not wish to add further length to an already lengthy paper. Our intention is to conduct a systematic literature review in addition to consulting experts in Uganda, to determine the best course of action for partnership engagement.

In order to address the example of community engagement, we referenced the below paper in the discussion. This article shows that community level interventions can be effective; however, it is not yet evaluated in the context of pediatric post-discharge care.

Reference: 28. Black RE, Taylor CE, Arole S, Bang A, Bhutta ZA, Chowdhury AMR, et al. Comprehensive review of the evidence regarding the effectiveness of community-based primary health care in improving maternal, neonatal and child health: 8. summary and recommendations of the Expert Panel. Journal of global health. 2017;7(1):010908.

Reviewer Comment 16: 423–425: HCW education and paradigm shift - elaborate on how this education and paradigm shift could be effectively integrated into the existing healthcare system. Are there specific training programmes, guidelines, or policies that could be used or developed?

Response 16: Thank you for highlighting that we should consider how this education and paradigm shift could be effectively integrated into the existing healthcare system. In response to the comment, we have added the suggestion that policy/guidance such as the Integrated Community Case Management (iCCM) needs to be improved to include children who were recently discharged from hospital (lines 439-442). The Smart Discharges research team is presently working with the Ministry of Health in Uganda to update iCCM to include “care of the discharged child”.

Reviewer Comment 17: 431-440: Male community member engagement - how can this engagement be developed, maintained and expanded in diverse cultural contexts?

Response 17: While we agree that male community member engagement is important and is indeed a key finding, expanding on how this can be achieved is beyond the scope of this paper. This topic is clearly very broad and we, unfortunately, do not have wish to detract from the main findings, which could result in a loss of readership. Future papers will focus on aspects related to strategy, which includes other aspects as well, such as building partnerships with community groups.

Reviewer Comment 18: 441-457: Financial constraints - i

---

## [Editor Report · Decision Letter 1]

1 Jul 2024

Caregivers’ and nurses’ perceptions of the Smart Discharges Program for children with sepsis in Uganda: A qualitative study

PONE-D-23-37812R1

Dear Dr. Wiens,

We’re pleased to inform you that your manuscript has been judged scientifically suitable for publication and will be formally accepted for publication once it meets all outstanding technical requirements.

Kind regards,

Chai-Eng Tan

Academic Editor

PLOS ONE
---

## [Editor Report · Acceptance letter]

12 Aug 2024

PONE-D-23-37812R1 

PLOS ONE

Dear Dr. Wiens, 

I'm pleased to inform you that your manuscript has been deemed suitable for publication in PLOS ONE. Congratulations! Your manuscript is now being handed over to our production team.

Kind regards, 

on behalf of

Dr. Chai-Eng Tan 

Academic Editor

PLOS ONE